# Benchmarking Foundation Models for Renal Lesion Stratification in CT

**Hartmut Häntze**[1,2,3] (iD)                    HARTMUT.HAENTZE@CHARITE.DE
[1] *Charité - Universitätsmedizin Berlin, Berlin, Germany*
[2] *Radboudumc, Nijmegen, The Netherlands*
[3] *Klinikum rechts der Isar, TUM University Hospital, Technical University of Munich, Munich, Germany*

**Sarah de Boer**[2] (iD)                         SARAH.DEBOER@RADBOUDUMC.NL
**Myrthe Buser**[2] (iD)                          MYRTHE.BUSER@RADBOUDUMC.NL
**Alessa Hering**[2] (iD)                         ALESSA.HERING@RADBOUDUMC.NL
**Bram van Ginneken**[2] (iD)             BRAM.VANGINNEKEN@RADBOUDUMC.NL
**Mathias Prokop**[2] (iD)                      MATHIAS.PROKOP@RADBOUDUMC.NL
**Jawed Nawabi**[1] (iD)                              JAWED.NAWABI@CHARITE.DE
**Sebastian Ziegelmayer**[3] (iD)                        S.ZIEGELMAYER@TUM.DE
**Lisa Adams**[3] (iD)                                      LISA.ADAMS@TUM.DE
**Keno Bressem**[3] (iD)                                 KENO.BRESSEM@TUM.DE

## Abstract

The rapid proliferation of open-source medical foundation models (FMs) raises a practical question: how well do their pre-trained representations transfer to clinically relevant but data-scarce classification tasks? We investigated this in a controlled benchmark of three medical FMs on renal lesion stratification, a multi-class problem, where training data is inherently scarce. Using a frozen feature-probing protocol, we compared FM embeddings against two established baselines: a radiomics classifier and a 3D ResNet-50 trained from scratch. Our results reveal two findings. First, FM performance matched the from scratch-trained ResNet, while drastically lowering the hardware demand. Second, the conventional radiomics baseline (AUC 0.88) outperformed all deep learning approaches (AUC [0.69, 0.77]), suggesting that, despite their potential to improve classification in data-scarce settings, medical FMs do not yet surpass established models for renal lesion stratification.

**Keywords:** Computed Tomography, Foundation Model, Renal Cell Carcinoma

## 1. Introduction

Medical foundation models (FMs) present a new methodological alternative in automated diagnosis. By leveraging large-scale pre-training, these models aim to extract robust feature representations from datasets that are otherwise too small for training deep neural networks from scratch (Schäfer et al., 2024). Renal lesion stratification offers a clinically relevant test case for this approach. While kidney cancer ranks as the 14th most frequent malignancy globally (IARC, 2021), the non-invasive characterization remains challenging. Deep learning approaches have demonstrated potential in addressing these gaps (Alhussaini et al., 2022; Uchida et al., 2022), even outperforming radiologists (Uhm et al., 2021). However, their broader clinical translation is hindered by the low prevalence of specific histological

subtypes. Addressing these performance limitations requires training methodologies capable of learning robust representations from such limited data. In this study, we therefore investigate the efficacy of three open-source FMs for renal lesion stratification primarily using a frozen feature-probing protocol. We benchmark these models against two established baselines: a handcrafted radiomics classifier and a 3D ResNet-50 neural network trained from scratch. All models are implemented within our open-source framework, RenalVision, available at https://github.com/hhaentze/RenalVision.

## 2. Methodology

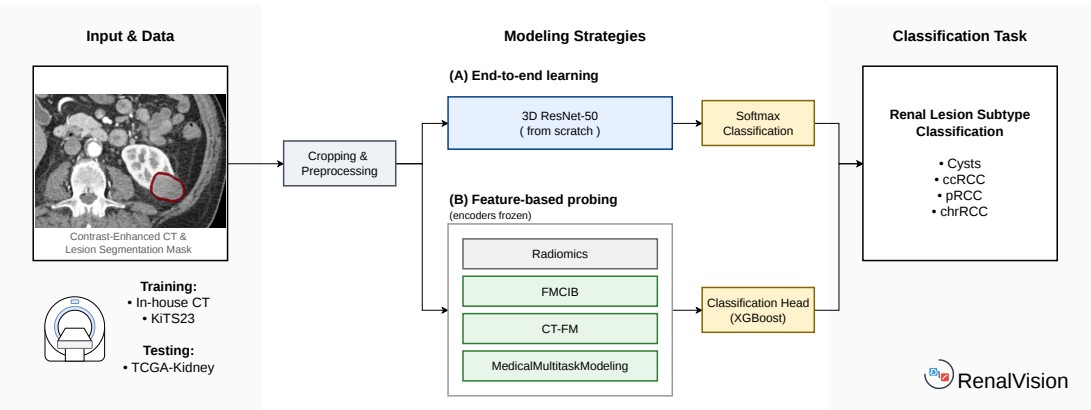

Figure 1: Overview of the processing pipeline

We categorized lesions into four distinct classes: cysts, clear cell renal cell carcinoma (ccRCC), papillary RCC (pRCC), and chromophobe RCC (chrRCC). The training dataset was compiled from two sources. First, we sampled histopathology confirmed renal lesions from an in-house contrast-enhanced CT dataset. Tumours and cysts were annotated by a student. This process yielded 1785 lesions from 796 patients (mean age $63.6 \pm 11.4$ years; 548 men). Second, we incorporated the KiTS23 dataset (Heller et al., 2023), consisting of annotated late arterial and nephrogenic phase CT scans from 483 patients. We excluded lesions with a diameter $< 1$ cm, resulting in 1069 lesions. The combined training dataset comprised 2854 lesions from 1279 patients. For external validation, three TCIA datasets (Akin et al., 2016; W. et al., 2016; Linehan et al., 2016) were automatically segmented (de Boer et al., 2025) and a subset manually verified by two students and a board-certified radiologist. This yielded 234 test lesions (94 cysts, 94 ccRCC, 28 pRCC, 18 chrRCC) from 101 patients.

We investigated three medical foundation models: FMCIB (Foundation Model for Cancer Imaging Biomarkers) (Pai et al., 2024), CT-FM (Vision Foundation Models for Computed Tomography) (Pai et al., 2025), and MMM (MedicalMultitaskModeling) (Schäfer et al., 2024; Nicke et al., 2025). Input volumes were all preprocessed according to the requirements of each model. After using the models for feature extraction we trained XG-Boost classifiers on top of them. Baselines included a default PyRadiomics model (Van Gri-

ethuysen et al., 2017) and a 3D-ResNet50 trained from scratch. The complete workflow is depicted in Figure 1. Performance was assessed using macro-average AUC and average precision (AP), testing statistical significance against baselines via simultaneous 95% CI bootstrapping.

## 3. Results & Discussion

Table 1: Performance metrics for the external test data (4 classes). Scores are reported as Mean (95% CI). Best results are marked in bold.

| Category | Model | AUC | AP |
|---|---|---|---|
| Baselines | ResNet-50 | 0.72 (0.68–0.77) | 0.45 (0.40–0.51) |
| | Radiomics | **0.88 (0.85–0.91)** | **0.64 (0.58–0.70)** |
| Foundation | CT-FM | 0.70 (0.65–0.74) | 0.43 (0.38–0.49) |
| Models | MMM | 0.77 (0.73–0.82) | 0.50 (0.44–0.56) |
| | FMCIB | 0.69 (0.64–0.74) | 0.40 (0.35–0.46) |

On external TCIA data the best-performing FM was MMM (AUC=0.77), followed by CT-FM (AUC=0.70) and FMCIB (AUC=0.69) (Table 1). These results are similar to the performance of the ResNet baseline (AUC=0.72), with no statistically significant differences. Noteworthy, this performance was achieved with a massive reduction in computational cost. While the FM-classifier combinations could all be trained on a CPU, the ResNet required hours of GPU-accelerated training. Nonetheless, the radiomics baseline significantly outperformed all FM approaches (AUC=0.88). This suggests that the information encoded in the latent representations of FMs is not yet as discriminative as explicitly engineered radiomics. The interpretability of radiomics further supports its clinical utility over opaque deep learning embeddings. Ultimately, while FMs provide an efficient, compute-light alternative to from-scratch deep learning for data-scarce renal lesion stratification, targeted radiomics remain the optimal approach; indicating that current generalist FM embeddings do not yet capture the fine-grained texture heterogeneity essential for precise histological subtyping.

## Acknowledgments

This work was funded by the European Union (grant no. 101079894). Views and opinions expressed are, however, those of the author(s) only and do not necessarily reflect those of the European Union or European Health and Digital Executive Agency (HADEA). Neither the European Union nor the granting authority can be held responsible for them. Much of the computation resources required for this research were performed on hardware provided by the Charité HPC cluster. The results published here are in whole or part based upon data generated by the TCGA Research Network: http://cancergenome.nih.gov.

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
