# OpenReview forum: "Benchmarking Foundation Models for Renal Lesion Stratification in CT"
_MIDL.io/2026/Short_Papers — MIDL 2026 - Short Papers Poster_

### Official Review · Reviewer_oMC1 · 2026-05-03
**Clear and informative benchmark of foundation models for data-scarce medical imaging**

**Rating:** 5
**Confidence:** 5

**Review:**

The paper is well written, clear and well motivated that addresses the question of whether medical foundation models can replace expensive from-scratch training in data-scarce clinical settings. Besides comparing FMs, the authors also compare with radiomics and a ResNet model trained from scratch. The use of confidence intervals for performance estimates is especially appreciated and adds credibility to the conclusions. While the work does not introduce a novel methodological contribution, it remains quite relevant for the short paper track, as it provides a grounded and insightful benchmark across multiple models and clarifies the strengths and limitations of FM embeddings for a challenging multi-class task.

**Summary:**

This paper presents a benchmark of medical foundation models (FMs) for renal lesion stratification in CT. The authors compare frozen FM embeddings with radiomics and a 3D ResNet trained from scratch and show that FMs achieve comparable performance to the ResNet while reducing training and hardware costs. Overall, the work offers practical insights into when FM embeddings are sufficient for complex clinical classification tasks.

**Strengths:**

- The authors propose a careful and fair benchmarking of multiple foundation models.
- Using foundation models can help reduce training cost while maintaining comparable performance with a ResNet model.
- Inclusion of confidence intervals strengthens statistical interpretation.
- The paper is very clear and well-motivated.

**Weaknesses:**

- Students performed a large portion of the annotations. However, since a senior radiologist reviewed the external validation dataset, its quality should be high.
- No analysis of inter-annotator variability or annotation noise is provided.

**Justification Of Rating:**

This work presents a well-executed benchmark that is highly relevant to the community. The paper provides clear, practical guidance on the trade-offs among foundation models, radiomics, and from-scratch training in a realistic, data-scarce setting. I believe it is a good fit for MIDL's short-paper track.

---

### Decision · Program_Chairs · 2026-05-08

Accept (Poster)